# Diagnostic Approach to Coccidioidomycosis in Solid Organ Transplant Recipients

**DOI:** 10.3390/jof9050513

**Published:** 2023-04-26

**Authors:** Tirdad T. Zangeneh, Mohanad M. Al-Obaidi

**Affiliations:** Division of Infectious Diseases, College of Medicine, University of Arizona, 1501 N Campbell Avenue, P.O. Box 245022, Tucson, AZ 85724, USA

**Keywords:** coccidioidomycosis, solid organ transplant, fungal diagnostic

## Abstract

Coccidioidomycosis is a fungal infection endemic in the southwestern United States, Mexico, and parts of Central and South America. While coccidioidomycosis is associated with mostly mild infections in the general population, it can lead to devastating infections in immunocompromised patients, including solid organ transplant (SOT) recipients. Early and accurate diagnosis is important in achieving better clinical outcomes in immunocompromised patients. However, the diagnosis of coccidioidomycosis in SOT recipients can be challenging due to the limitations of diagnostic methods including cultures, serology, and other tests in providing a timely and accurate diagnosis. In this review, we will discuss the available diagnostic modalities and approaches when evaluating SOT recipients with coccidioidomycosis, from the use of conventional culture methods to serologic and molecular testing. Additionally, we will discuss the role of early diagnosis in assisting with the administration of effective antifungal therapy to reduce infectious complications. Finally, we will discuss ways to improve the performance of coccidioidomycosis diagnostic methods in SOT recipients with an option for a combined testing approach.

## 1. Introduction

Coccidioidomycosis (“valley fever”) is a fungal infection caused by two genetically distinct *Coccidioides* species (*Coccidioides immitis* and *Coccidioides posadasii*), which are soil-inhabiting molds endemic in the southwestern United States (US), Mexico, and parts of Central and South America. The rate of reported coccidioidomycosis infections has been increasing over recent decades. This is partly thought to be secondary to climate change, shifts in population with increased residence and travel to endemic areas, increased testing, and the availability of improved diagnostic tests [1]. While most infections are asymptomatic or mild, moderate to severe infection carries high morbidity and mortality, especially in immunocompromised patients, including in solid organ transplant (SOT) recipients [2,3,4]. It is thought that less than 5% of coccidioidomycosis cases in the general population progress to disseminated disease. However, several studies have reported dissemination rates as high as 30%, with mortality rates of 13% to 29% among SOT recipients [5,6,7]. Unfortunately, infections tend to be missed or underdiagnosed, even in endemic regions where greater familiarity with the manifestations of this infection is expected. In a recent study, it was reported that Arizona clinicians’ inability to diagnose coccidioidomycosis in ambulatory settings led to the majority (73%) of new cases being identified during hospitalizations [8]. These missed cases can lead to severe coccidioidomycosis complications and death, especially among SOT recipients, who are at a greater risk of experiencing severe disease. Therefore, the current clinical practice guidelines for SOT recipients recommend the administration of antifungal prophylaxis during the post-transplant period [9]. In certain circumstances, SOT recipients with a history of past coccidioidomycosis, de novo coccidioidomycosis, and/or donor-derived infection may need lifelong antifungal prophylaxis to prevent severe infection. These recommendations are based on observations that close to 66% of coccidioidomycosis cases occur during the first year of transplant, with the majority occurring within six months after transplantation [10]. Despite all these measures, some cases of de novo coccidioidomycosis and reactivated infections can be missed. Hence, the timely and accurate diagnosis of coccidioidomycosis in SOT recipients, while critically important, can be challenging and delayed [11]. Serologic testing is the mainstay of clinical diagnosis; however, this testing modality is less sensitive in immunocompromised patients [12,13]. This is especially concerning as SOT recipients are at an increased risk of coccidioidomycosis-related mortality. Therefore, early diagnosis is crucial in providing the necessary information for clinicians to assist with the initiation of appropriate antifungal therapy. This is because post-transplant mortality has been reported to be as high as 63% in SOT recipients [4], while donor-derived infections have reported mortality rates of 28% [13,14]. While most of the literature has focused on reactivation or donor-derived cases of coccidioidomycosis in SOT recipients, we previously reported that 40.7% of SOT recipients experienced possible de novo pulmonary coccidioidomycosis as well, with 5.5% having an extrapulmonary disease, resulting in a mortality rate of 1.1% [2].

In addition to serologic testing, cultures and pathology have historically provided a definitive diagnosis. However, results of diagnostic tests are usually not readily available at the time of clinical presentation, resulting in a delay in coccidioidomycosis diagnosis. These delays often lead to the late recognition of more severe and protracted infections and hinder the initiation of timely antifungal therapy. Newer diagnostic methods such as galactomannan antigen testing, (1–3) Beta-d-glucan (BDG), and polymerase chain reaction (PCR) may be useful additional tests when diagnosing SOT recipients with suspected coccidioidomycosis. However, some of these tests, including BDG, are not specific to coccidioidomycosis and may not eliminate the diagnosis of concurrent fungal infections in immunocompromised patients. [15,16]. Therefore, it is our goal to provide further guidance when diagnosing suspected coccidioidomycosis cases in SOT recipients. We will also discuss the different diagnostic methods while highlighting their limitations.

## 2. Clinical Presentation

The clinical presentation of coccidioidomycosis ranges from mild symptoms to severe disease, usually represented by acute pulmonary symptoms, such as cough, pleuritic chest pain, dyspnea, and fever. Some of the pulmonary presentations mimic those experienced with bacterial pneumonia. Other presentations may be associated with extrathoracic and disseminated infection, mimicking autoimmune diseases, malignancy, and other non-infectious etiologies, with symptoms ranging from constitutional, such as fever, weight loss, night sweats, and headache, to neurological deficits, or cutaneous, osteoarticular, or genitourinary involvement [17]. While most individuals living in the endemic region may be exposed to coccidioidomycosis, most develop mild or no symptoms and may not even be aware that they have had coccidioidomycosis [18]. However, abnormalities affecting the immune system are associated with severe diseases. Such immunologic conditions can be inherited [19] and/or acquired, as in the case of SOT recipients treated with immunosuppressive medications used to prevent allograft rejection [4]. The clinical presentation of coccidioidomycosis in SOT recipients can be more challenging due to the likelihood of patients presenting with severe, protracted, disseminated, and atypical manifestations mistaken for bacterial and other fungal infections. Indeed, coccidioidomycosis has been identified as a cause of community-acquired pneumonia (CAP) in Arizona, where in patients with CAP, close to 29% had serologic positivity for coccidioidomycosis in one study [17]. Therefore, a high level of clinical suspicion is required when evaluating immunosuppressed patients who have been exposed to endemic regions and/or received organs from donors from the endemic areas. More importantly, a timely diagnosis is imperative to providing the proper therapeutic interventions. Due to the endemicity of this infection, the diagnosis and management of coccidioidomycosis require a multidisciplinary team of experts comprising of infectious diseases specialists, pathologists, microbiologists, pulmonologists, pharmacists, radiologists, surgeons, and other specialists familiar with the presentations and manifestations of this endemic fungal infection.

## 3. Fungal Cultures and Pathology

Traditionally, fungal cultures and histopathology performed on collected specimens are required to achieve a definitive diagnosis. However, there are several limitations to these diagnostic methods in general, as diagnostic limitations affect immunocompromised patients more specifically. Coccidioidomycosis can be diagnosed via histopathology through the identification of *Coccidioides* spp. spherules in Grocott methenamine silver (GMS), periodic acid Schiff (PAS), or hematoxylin-eosin (H & E) staining [20]. However, the histopathological diagnostic yield depends on several factors, including the appropriate and adequate collection of specimens, which may reduce the sensitivity of this method. In cases where identification occurs after the growth of cultures obtained through bronchioalveolar lavage (BAL), less than half of the samples are positive in cytopathology [21].

Overall, specimens evaluated through pathology and microbiology can be limited by their lower sensitivity, difficulty in obtaining these specimens, and the longer time required to yield results. Moreover, to isolate *Coccidioides* spp. using culture methods, a special microbiology laboratory is needed as the identification process can pose an occupational risk to the laboratory staff. This is due to possible airborne or direct inoculation exposure associated with the occupational hazard related to the identification of these organisms [22,23]. *Coccidioides* spp. are usually isolated in fungal media such as brain–heart infusion agar, potato dextrose or potato flake agar, and Sabouraud dextrose agar, and sometimes in bacterial media (sheep blood and chocolate agars) [20]. *Coccidioides* spp. can be identified using different methods after growth in media [24]. Depending on colony maturity, microscopy can be used to identify *Coccidioides* spp. by the morphology of the mycelial form. Several commercial labs have developed deoxyribonucleic acid (DNA) probes for the identification of *Coccidioides* spp. [25], which can offer a more specific and possibly faster method for identification. Recently, matrix-assisted laser desorption/ionization–time-of-flight (MALDI-TOF) mass spectrometry (MS) has been shown to be a promising tool in identifying various fungal pathogens, including dimorphic fungi [26]. Despite the challenges accompanying microbiological and histopathological methods of identification, obtaining adequate samples of the anatomically involved sites is important in achieving a definite diagnosis and for ruling out other infectious diseases.

### Coccidioides Serologic Tests

Coccidioidomycosis has historically been diagnosed through culture and pathology, but with the development of serologic tests, these methods have become the alternative method for diagnosing coccidioidomycosis [27]. These tests rely on the immune responses, and it may take 2–4 weeks after acute infection for individuals to develop detectable antibodies. The current medical consensus among experts for diagnosing coccidioidomycosis places the emphasis on serologic methods, used in the correct clinical context and with the appropriate epidemiological information [27]. *Coccidioides* serologic tests are available through different specialized commercial labs and can have a range of sensitivities (Table 1). Most commercial laboratory tests rely on enzyme immunoassays (EIAs), which are qualitative or semiquantitative assays used for the diagnosis of coccidioidomycosis. The *Coccidioides* EIA has the highest sensitivity, but it is prone to false positive results, especially in cases of an isolated IgM. In one study, the rate of false positive IgM was reported to be as high as 82%, contributing to further complexity when diagnosing patients [28]. Therefore, it is recommended that a positive EIA test is followed by a confirmatory immunodiffusion (ID) test, which is associated with a lower sensitivity of 30–60% a high reported specificity [29,30,31]. However, ID utility in SOT is not well evaluated. Similarly, while a lateral flow assay (LFA) showed low sensitivity in detecting coccidioidomycosis in immunocompetent patients early in the disease, the utility of such testing has not been well evaluated in SOT recipients [29].

Another challenging issue with the use of *Coccidioides* EIA is the interpretation of an indeterminate test result, which is defined according to different manufacturer instructions and is considered the same as a nonreactive result by most experts. However, it is not uncommon for our SOT recipients to have an indeterminate test result from one EIA assay followed by a positive test result with a different assay. In addition, there are SOT recipients with indeterminate EIA test results who later develop severe infections and test positive with a subsequent alternate diagnostic test. Considering these indeterminate EIA test results as negative is an acceptable practice when evaluating asymptomatic immunocompetent patients. However, employing such an approach may have dire clinical consequences in SOT recipients and other immunosuppressed patients. To highlight the importance of this issue, in a study of liver transplant recipients in Arizona, the authors reported that of the five patients with indeterminate serologic results prior to transplantation who had not received antifungal prophylaxis, one patient died of disseminated coccidioidomycosis shortly after transplantation [9]. In our center, SOT recipients with presumed coccidioidomycosis and an indeterminate EIA test result undergo repeat testing after 2–4 weeks. These SOT candidates and recipients are closely monitored for the development of active infection during follow-up visits. More recently, patients with indeterminate EIAs have also undergone alternative MiraVista anti-*Coccidioides* EIA testing [39]. The results of this diagnostic strategy at our center will be formally studied and reported on in the near future. Currently, there are several available EIAs in commercial laboratories with varying diagnostic performances. As such, we reported that the MiraVista anti-*Coccidioides* EIA might have a higher sensitivity in diagnosing immunosuppressed hosts with coccidioidomycosis. However, our study included a small subset of a heterogenous immunosuppressed population. Therefore, further studies are required to assess the performance of the MiraVista anti-*Coccidioides* EIA in SOT patients [29]. *Coccidioides* EIAs are often employed in the serologic testing of coccidioidomycosis and can provide a rapid and uncomplicated diagnostic tool for clinicians. 

The *Coccidioides* complement fixation (CF) antibody test is mainly used prognostically through the evaluation of serum and body fluids. CF testing is especially useful for cerebrospinal fluid (CSF) samples in the diagnosis of meningitis [30]. This test, which is used mainly for evaluating disease severity, prognosis, and response to therapy, measures IgG antibody (qualitative and semiquantitative) binding to complements and thereby inhibiting the lysis of foreign red blood cells [31]. Many patients with detectable CF titers undergo serial testing as a means of monitoring their response to antifungal therapy.

However, the reliance on serologic testing in immune responses makes it vulnerable to diagnostic challenges, particularly among immunocompromised individuals. This can result in delayed diagnosis in patients with coccidioidomycosis, including those with SOTs, leading to the progression of their infection and potentially grave clinical outcomes. Despite the acknowledged limitations of serologic testing in this population, the search for alternative diagnostic methods is ongoing to establish reliable diagnostic strategies for coccidioidomycosis in SOT recipients.

## 4. Fungal Antigen Markers

*Coccidioides* spp. cell wall contains several glycoproteins and polysaccharides that have been found to be potential targets for different diagnostic assays. The galactomannan glycoprotein is one of these studied cell wall components utilized in the diagnosis of different fungal infections [40]. This antigen was found to be detected in the blood, urine, and CSF of patients with coccidioidomycosis with different reported sensitivities [34]. MiraVista laboratory developed a *Coccidioides* GM test that showed high sensitivity and specificity in the cases of *Coccidioides* meningitis, and a moderate level of sensitivity and specificity in other disseminated infections [35,41]. Among immunosuppressed patients with severe or disseminated disease, the sensitivity of the *Coccidioides* galactomannan antigen was 75%. However, when serologic testing was combined with GM antigen testing, the sensitivity was higher (93%) than when either test was performed alone [40]. 

The BDG is another antigen marker commonly tested in sera samples and known to be associated with a myriad of invasive fungal infections [42,43]. In a study of immunocompetent patients residing in the coccidioidomycosis endemic region, BDG was reported to detect early severe disease with a sensitivity of 44% and specificity of 92% [32]. We previously reported a high rate of agreement (90%) between the *Coccidioides* GM antigen and BDG in the sera of patients with coccidioidomycosis [44]. In addition to testing BDG in sera samples, CSF BDG has shown promise in diagnosing *Coccidioides* meningitis with a reported sensitivity of 96% and specificity of 82% [45]. However, the study utilized a BDG cut-off threshold of 31 pg/mL as a positive test, which is not the widely accepted cut-off value, and after increasing the cut-off value to >100 pg/mL, a higher specificity was achieved but with a lower sensitivity. Serum BDG testing may be a useful diagnostic tool for coccidioidomycosis in SOT recipients. We recently investigated the role of serum BDG in a cohort of hospitalized immunosuppressed patients with culture-proven coccidioidomycosis. We found that serum BDG had a sensitivity of approximately 50%, which was similar to the performance of serologic assays used in the same patient group. However, when we combined the use of both serologic tests and BDG, the diagnostic rate for coccidioidomycosis potentially increased to 82%. Our study utilized a serum BDG level of 80 pg/mL as the cut-off value for a positive test result. It is important to note that we did not evaluate the specificity of serum BDG in our investigation. These findings underscore the potential utility of employing multiple diagnostic modalities to improve the sensitivity of coccidioidomycosis diagnosis in SOT recipients [33]. Nevertheless, BDG assay results should be interpreted with caution, since it is known that BDG may be expressed by fungal pathogens other than coccidioidomycosis in immunosuppressed patients. Additionally, serum beta-D-glucan (BDG) testing may yield false positive results in patients undergoing hemodialysis with cellulose membranes and those receiving immunoglobulin products, albumin, or other blood products [46,47]. Despite this limitation, a positive BDG test result in the appropriate clinical context may serve as an important initial indicator to suggest coccidioidomycosis and prompt early antifungal therapy to prevent potential clinical complications. 

## 5. Fungal Molecular Tests

The timely diagnosis of coccidioidomycosis can be hampered by the delay in isolating *Coccidioides* spp. from fungal cultures and/or through histopathologic identification. Therefore, the utilization of different concurrent methods as mentioned above may provide a faster result. Another diagnostic tool is the use of rapid molecular tests, which may be applied to *Coccidioides* spp. isolated from fungal cultures from tissue or fluid samples. However, stand-alone molecular testing can be a powerful tool in the diagnosis of coccidioidomycosis, especially in SOT recipients. The *Coccidioides* PCR test is promising as it can provide a rapid and specific result, but like culture and pathology, its sensitivity and specificity are dependent on the quality of samples obtained [36,37,38]. A study validating the PCR test using a GeneSTAT *Coccidioides* assay on respiratory samples positive for *Coccidioides* spp. by culture showed a sensitivity of 100% and specificity of 95% [37]. However, this study included samples that were positive by culture for *Coccidioides* spp. In another study evaluating *Coccidioides* spp. PCR testing while using a different assay, the sensitivity of testing was lower (74%) when the diagnosis was made in patients with positive serologic testing or cultures [36], reflecting the importance of the tested sample quality and the level of DNA detection. Currently available diagnostic tests for coccidioidomycosis are predominantly pathogen-specific and involve the amplification of sections of *Coccidioides* spp. DNA. The emergence of next-generation sequencing (NGS) technology has revolutionized the diagnosis of infectious diseases, including coccidioidomycosis [48]. By leveraging NGS, different fungal pathogens including *Coccidioides* spp. can be accurately identified in patients with coccidioidomycosis [15,16]. This approach is particularly useful in SOT recipients, who are at a heightened risk for the development of multiple simultaneous infections. Therefore, NGS holds great promise as a powerful diagnostic tool for coccidioidomycosis in immunocompromised patients. 

## 6. Combined Diagnostic Testing

The diagnosis of coccidioidomycosis in SOT patients can be challenging and requires a high level of clinical suspicion in the correct clinical presentation and epidemiological milieus. Despite the advances in diagnostic testing, the results of these tests can be delayed and lack sensitivity or specificity in achieving the correct diagnosis in high-risk populations. Blair et al. previously highlighted the utility of a combined diagnostic approach in SOT recipients with coccidioidomycosis. In their study, they reported that the positivity of any single serologic test ranged from 21% to 56% compared with 77% when utilizing a battery of serologic tests including EIA, ID, and CF. Repeating serologic tests a month later increased their positive test findings to 92%, and the authors recommended that the use of multiple test modalities and repeat testing may increase the sensitivity of diagnostic assays in SOT recipients [11]. Similarly, as mentioned above, combining serologic and antigenic markers can result in a higher diagnostic sensitivity in immunosuppressed patients [33,34]. Therefore, we follow such an approach when evaluating SOT patients with a high clinical suspicion of having moderate to severe coccidioidomycosis. In the context of presumed infection, the primary consideration is the epidemiologic risks for SOT recipients and the importance of considering the donor’s status in the evaluation process. However, this type of information may not always be available and poses a challenge for patients who may have resided or traveled to endemic regions in the past. In addition, the timing, clinical presentation, and severity of infection are all important aspects of diagnosis. In such cases, utilizing a combination of diagnostic modalities including cultures from blood and affected site(s), pathology, serology, GM antigen, PCR, and BDG tests can aid the diagnosis (Figure 1A). The use of various imaging modalities and cerebral spinal fluid (CSF) analysis in the correct clinical context are additional tools utilized in the diagnosis of coccidioidomycosis. The diagnostic approach may also be different when approaching asymptomatic SOT recipients with suspected coccidioidomycosis in the clinic or hospital setting (Figure 1B). However, as recommended by the Infectious Disease Society of America (IDSA) guidelines, early initiation of appropriate antifungal therapy in SOT recipients exhibiting rapidly progressive and concerning clinical findings suggesting coccidioidomycosis may result in decreased morbidity and mortality [30].

## 7. Future Directions

The delayed-type hypersensitivity (DTH) reaction in coccidioidomycosis, tested through intradermal inoculation, thereby stimulating a cellular immune response commonly referred to as a skin test (CST) conversion, was originally available until the 1990s [49]. In 2014, a newly formulated *Coccidioides* skin test, developed as a spherule-derived antigen preparation (Spherusol, Nielsen Biosciences, San Diego, CA, USA), became commercially available for clinical and epidemiologic use [49,50]. Following its approval in adults with a history of pulmonary coccidioidomycosis, Blair et al. presented their data for *Coccidioides* (spherulin) CST, where 72% of patients with pulmonary coccidioidomycosis had a positive result. They also noted multiple technical factors impacting the results, including local site reactions, anergy, and dependence on technical expertise in the correct interpretation of the results [51]. As such, the use of *Coccidioides* (spherulin) CST fell out of favor for diagnostic purposes. Recently there has been growing interest in the development of interferon gamma (IFN-γ) release assays in diagnosing and evaluating immune responses in patients with coccidioidomycosis. Like the diagnosis of latent tuberculosis, the IFN-γ release assay utilizing whole blood incubated with *coccidioidal* antigen was studied by Ampel et al. [52]. The purpose of this study was to assess cellular immunity response in coccidioidomycosis and IFN-γ production in correlation with active coccidioidomycosis. Furthermore, they reported that an elevated IFN-γ concentration was associated with a lower clinical severity score. Although the authors noted a possible utility for diagnosing coccidioidomycosis, the low sensitivity and specificity could not confidently predict the occurrence of disseminated or chronic infection [51]. More recently, Ampel et al. evaluated the ex vivo release of several inflammatory proteins in patients with acute pulmonary coccidioidomycosis. In patients with pulmonary coccidioidomycosis, several cytokines were increased in high concentrations, including granulocyte-macrophage colony-stimulating factor (GM-CSF), interleukin-1β (IL-1β), interferon gamma (IFN-γ), IL-2, IL-13, and tumor necrosis factor alpha (TNF-α) [53]. These findings may have a significant implication in the diagnosis of acute pulmonary coccidioidomycosis in the near future and may help clinicians understand the course and the immune response of coccidioidomycosis in the SOT population.

Other investigational approaches in diagnosing coccidioidomycosis include novel methods borrowing from the current understanding of other fungal diagnostic modalities. One approach is through identifying volatile organic compounds (VOC), shown to be promising in the identification of other fungal infections such as invasive aspergillosis [54]. In a report, VOC was shown to detect *Coccidioides* spp. and assist in the development of a breath test to be implemented in clinical settings [55]. Another innovative method requiring further investigation is the use of targeted liquid chromatography–tandem mass spectrometry-based metabolic profiling, capable of identifying 207 plasma metabolites and 231 urinary metabolites detected in *Coccidioides* spp. metabolic pathways. When samples were collected from 48 patients with coccidioidomycosis and 99 controls, this method had a 94.4% sensitivity and 97.6% specificity for plasma and 89.7% sensitivity and 88.1% specificity for urine metabolites [56]. Finally, imaging modalities such as computer tomography scans, including positron-emitted tomography, may help identify pulmonary and disseminated coccidioidomycosis-associated lesions [57]. This information can be fed into a neural network to provide a rapid and accurate diagnosis. This approach was shown to be promising in a non-human study involving radiographs in dogs with coccidioidomycosis [58]. These novel diagnostic modalities require further evaluation, especially in SOT recipients, who require rapid, sensitive, and accurate tests to assist in prompt diagnosis and treatment.

## 8. Conclusions

As coccidioidomycosis is associated with high morbidity and mortality in SOT recipients, the awareness of diagnostic modalities and limitations in this population is critical when evaluating immunocompromised patients with suspected moderate to severe coccidioidomycosis. As such, we recommend an approach utilizing multiple testing modalities, including cultures, pathology, serology, antigen, PCR, and BDG tests, and imaging in SOT recipients with life-threatening infections (Figure 1A). When SOT recipients present with signs and symptoms that are concerning for moderate to severe coccidioidomycosis, we recommend initiating an extensive workup. This includes *Coccidioides* serologic testing, which should be repeated in 2–4 weeks to increase sensitivity. Additionally, we combine initial work up with serum BDG, and blood and urine *Coccidioides* GM antigen tests, with specific sampling of affected sites for cultures, pathology, and other specific diagnostic tests, and the latter for more specific evaluation. For patients with pulmonary involvement, we recommend the addition of induced sputum, or preferably BAL or biopsy for both fungus culture and histopathology evaluation. Samples may also be submitted for PCR testing. For cases with pleural involvement, thoracocentesis for pleural fluid or lung biopsy analyzed for fungus culture and *Coccidioides* CF titers and additional testing are recommended. Lastly, for suspected extrapulmonary involvement, we recommend sampling the involved site(s) (bone, joint, soft tissue, etc.) and sending tissue for fungal culture, histopathology, and PCR testing if possible. For suspected central nervous system infections including meningitis, we recommend obtaining imaging and lumbar puncture for CSF analysis using fungus culture, *Coccidioides* CF titer, *Coccidioides* GM, and BDG tests. When utilizing such an approach, attention to diagnostic stewardship, diagnostic cost, and inpatient length of stay is important. It is also important to note that such an approach is not recommended for all SOT recipients and candidates. Many SOT recipients and candidates may be asymptomatic at the time of evaluation and suspected to have coccidioidomycosis based on a positive or indeterminate serologic test. As such, these patients can undergo testing strategies similar to those employed in the diagnosis of immunocompetent patients with careful and frequent monitoring and a low threshold for initiating appropriate antifungal therapy when necessary. It is important to note that utilizing a combined diagnostic approach requires familiarity with the strengths and limitations of each testing modality. Future diagnostic studies are needed to investigate novel methods while including SOT recipients. These future studies require evaluating a combined approach to maximize the sensitivity and specificity of such diagnostic modalities. The development of newer and improved tests, along with the careful selection of patients at the highest risk for complications, may help offset the cost and risks associated with expanded diagnostic testing. In conclusion, while the diagnosis of coccidioidomycosis in SOT recipients can be challenging, early recognition can aid the workup through the involvement of necessary clinical, imaging, and laboratory experts to assist with diagnosis and management in a timely manner to improve outcomes.

## Figures and Tables

**Figure 1 jof-09-00513-f001:**
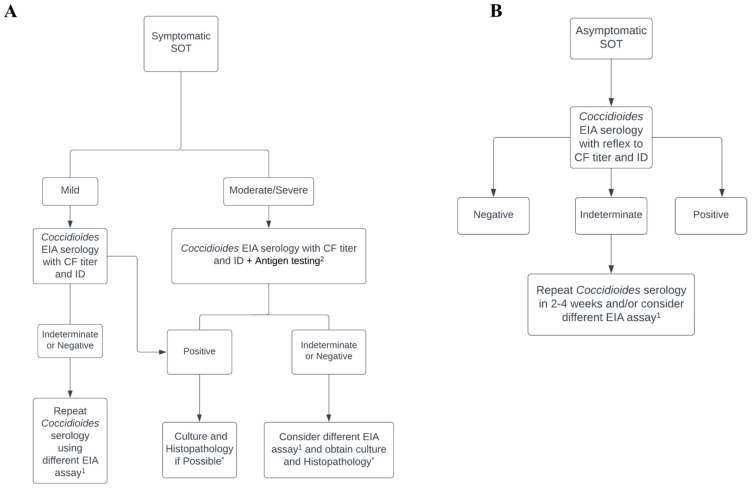
Diagnostic screening and testing for coccidioidomycosis in SOT candidates and recipients. Diagnostic approach to coccidioidomycosis in SOT recipients (**A**). Screening for Coccidioidomycosis in SOT recipients and candidates (**B**). SOT, solid organ transplant; EIA, enzyme immunoassay; ID, immunodiffusion; CF, complement fixation. ^1^ MiraVists *Coccidioides* EIA assay is utilized at our center as an alternative EIA test. ^2^ Antigen testing for Coccidioides Galactomannan and (1–3) Beta-d-glucan assay. * Samples for histopathology and culture of the appropriate site, and in severe cases, fungal blood cultures should be collected as well.

**Table 1 jof-09-00513-t001:** The spectrum of sensitivity and specificity of coccidioidomycosis diagnostic tests in SOT recipients.

Diagnostic Test	Sensitivity	Specificity	Comments	Refernces
Serology			Different enzyme immunoassays have varying degrees of reported sensitivity and specificity.	[12,29,30,31]
Miravista *	87%	90%		
Meridian	40–70%	95%		
IMMY	40–70%	95%		
(1–3) Beta-d-glucan (serum)	44–57%	Unknown	Serum (1–3) Beta-d-glucan is not specific to coccidioidomycosis.	[32,33]
*Coccidioides* Galactomannan Antigen (serum)	50%	95%	*Coccidioides* Galactomannan antigen has high sensitivity in CSF but lower sensitivity in blood and urine samples, except for cases of severe disease.	[34,35]
*Coccidioides* spp. culture	50%	100%	Sensitivity of the fungal culture is dependent on the obtained sample and microbiology laboratory.	[22]
*Coccidioides* spp. pathology	50%	100%	Sensitivity of the histopathology is dependent on the obtained sample, burden of disease, and pathologist expertise.	[22,23]
*Coccidioides* spp. PCR	70–90%	100%	Very promising technology but lacks real-world data and is likely sample-dependent.	[36,37,38]

PCR, polymerase chain reaction; CSF, cerebrospinal fluid; IMMY, Immuno Mycologics, Inc, Norman, OK, USA. * Based on a study that evaluated the sensitivity and specificity of antibody tests in various immunosuppressed patients.

## Data Availability

Not applicable.

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
