# Peer review of "Diagnostic Approach to Coccidioidomycosis in Solid Organ Transplant Recipients"

_jof, 2023, doi:10.3390/jof9050513_

Round 1

Reviewer 1 Report

I appreciate the opportunity to review this manuscript; it is well-written and engaging. The authors of this manuscript conducted a review of Coccidioidomycosis diagnosis in solid organ transplant recipients.

 I just have a few comments.

The subtitle “Clinical presentation” could include the pulmonary and extrapulmonary infection symptoms.

Subtitle “Fungal cultures and pathology”, 

Lines 119 to 125 should be moved to line 112.

Line 133:  You should use tests instead of markers.

Line 142: Enzyme Immunoassays (EIAs).

Line 147: Use ID rather than IMDF.

Lines 185 – 186: You should move these lines to line 177

You should include information about Immunodifusion and lateral flow assay.

Do Miravista, Meridian, and IMMY detect antigens or antibodies?

Table 1: 

Use percentages instead of +++

You should include CF and IMDF

you should include Immuno Mycologics inc in the figure’s legend.

A column titled "Sample(s)" might be interesting to include.

Please, include another column for the References

Subtitle “Fungal antigen markers

Line 214: ml to mL

Subtitle “Fungal molecular tests

Line 249: Include the sensitivity value, please.

Subtitle “Combined diagnostics testing

Line 286: Infectious Disease Society of America (IDSA)

Figure 1: ID to IMDF

Overall comments:

You should include information about the best moment to carry out each test to improve specificity. In other words, at what stage of the infection will the specificity of an EAI, CF, skin test, PCR, etc. be better?

Please unify the font of the letter.

Reviewer 2 Report

Very interesting article about coccidioidomycosis. Review with many bibliographical references. The authors focus on SOT recipients but explain very well the clinical and diagnostic aspects in a more general way.

There are many acronyms used throughout the text, please consider clarifying them all.

The authors should add a section on available treatments for this fungus or discuss treatments.

This review would be much more pleasant to read if it were more illustrated.
